# Biphasic Fermentation of *Trapa bispinosa* Shells by *Ganoderma sinense* and Characterization of Its Polysaccharides and Alcoholic Extract and Analysis of Their Bioactivity

**DOI:** 10.3390/molecules29061238

**Published:** 2024-03-11

**Authors:** Xiaoyan Sun, Qiuqi Lei, Qinyi Chen, Dandan Song, Min Zhou, Hongxun Wang, Limei Wang

**Affiliations:** 1College of Life Science and Technology, Wuhan Polytechnic University, Wuhan 430023, China; s514150539@163.com (X.S.); 18278911084@163.com (Q.C.); 13972457283@163.com (D.S.); wanghongxunhust@163.com (H.W.); 2College of Food Science and Engineering, Wuhan Polytechnic University, Wuhan 430023, China; lqq19970706@163.com (Q.L.); mzhou268@163.com (M.Z.)

**Keywords:** *Trapa bispinosa* shell, *Ganoderma sinense*, polysaccharides, UPLC-QTOF-MS/MS, immune activity, antioxidant activity

## Abstract

Background: *Trapa bispinosa* shells (TBs) and its flesh (TBf) have been recognized for their medicinal properties, including antioxidant, antitumor, and immunomodulatory effects. Despite these benefits, TBs are often discarded as waste material, and their applications remain to be further explored. Methods: In this study, we optimized the solid-state fermentation process of *Ganoderma sinense* (GS) with TBs using a response surface experiment methodology to obtain the fermented production with the highest water extract rate and DPPH free radical scavenging activity. We prepared and characterized pre-fermentation purified polysaccharides (P1) and post-fermentation purified polysaccharides (P2). Alcoholic extracts before (AE1) and after (AE2) fermentation were analyzed for active components such as polyphenols and flavonoids using UPLC-QTOF-MS/MS (ultra-performance liquid chromatography–quadrupole time-of-flight tandem mass spectrometry). Mouse macrophages (RAW 264.7) were employed to compare the immune-stimulating ability of polysaccharides and the antioxidant activity of AE1 and AE2. Results: Optimal fermentation conditions comprised a duration of 2 days, a temperature of 14 °C, and a humidity of 77%. The peak water extract yield and DPPH free radical scavenging rate of the water extract from TBs fermented by GS were observed under these conditions. The enhanced activity may be attributed to changes in the polysaccharide structure and the components of the alcoholic extract. The P2 treatment group indicated more secretion of RAW 264.7 cells of NO, iNOS, IL-2, IL-10, and TNF-α than P1, which shows that the polysaccharides demonstrated increased immune-stimulating ability, with their effect linked to the NF-кB pathway. Moreover, the results of the AE2 treatment group indicated that secretion of RAW 264.7 cells of T-AOC and T-SOD increased and MDA decreased, which shows that the alcoholic extract demonstrated enhanced antioxidant activity, with its effect linked to the Nrf2/Keap1-ARE pathway. Conclusions: Biphasic fermentation of *Trapa bispinosa* shells by *Ganoderma sinense* could change the composition and structure of the polysaccharides and the composition of the alcoholic extract, which could increase the products’ immunomodulatory and antioxidant activity.

## 1. Introduction

*Trapa bispinosa Roxb*. is an annual aquatic herb, also known as water chestnut. It has a rich historical cultivation as a traditional fruit and vegetable, spanning thousands of years [1]. The *Trapa bispinosa* shells (TBs) are rich in flavonoids [2,3], polyphenols [4], alkaloids [5], and various active ingredients with antitumor [6], antioxidant [7], antibacterial [8], and immune regulatory functions [9]. Notably, discarded TBs have been the focus of extensive research in recent years. Huang et al. [10] utilized 95% ethanol extraction at room temperature for 72 h to extract TBs, followed by ethyl acetate extraction of the crude extract thrice. This process yielded nine phenolic components, including 2,6-3-O-gallic acyl-beta-,3,6 d-glucose, 1-3-O-gallic acyl-beta-D-glucose, 1,2,3,4-4-O-gallic acyl-beta-D-glucose, and 1,2,3,4,6-five-O-gallic acyl-beta-D-glucose. Yu et al. [11] detected abundant phenolic compounds such as protocatechuic acid, gallic acid, and ferulic acid in both aqueous and alcoholic extracts obtained from different TBs varieties, with the alcoholic extraction demonstrating superior antioxidant effects compared to water extraction. Chiang et al. [12] evaluated TBs’ antioxidant effect by examining the impact of different milling powers on flavonoid and polyphenol extraction rates and DPPH free radical scavenging rates, concluding that TBs exhibited significant antioxidant activity correlated with flavonoid and polyphenol content. Xia et al. [13] showed the antioxidant, antiproliferative activity of the extract of *Trapa bispinosa* leaves; the alcoholic extract was absorbed by macroporous resin (D101) and eluted with different ratios of methanol–water and water; Fr.9 (eluted with 80% methanol) showed the highest antiproliferative activity in relation to HepG2, SSMC-7721, Hela, and A549 cells, and showed high antioxidant activity in relation to A549 cells. Ramsankar Sarkar [14] et al.’s study mentioned that the polysaccharides of *Trapa bispinosa* showed high splenocyte, thymocyte and macrophage activation, and antioxidant activity.

As a significant edible and medicinal fungus, *Ganoderma sinense* has been employed in traditional Chinese medicine for over 2000 years [15]. Current research on *Ganoderma sinense* composition has revealed richness in triterpenoid polysaccharides [16], amino acids, alkaloids, and other compounds [17]. Wu et al. [18] identified the connection between 1,3-β-D-glycoside and 1,4-α-D-galactoside in *Ganoderma lucidum* and *Ganoderma sinense* through mapping and comparison with existing saccharides. Lin Z [19] investigated the anticancer activity of *Ganoderma lucidum* polysaccharide, discovering its ability to promote tumor cell apoptosis and enhance the body’s immune activity. Hasnat et al. [20] demonstrated that *Ganoderma* triterpenoids could effectively inhibit the expression of Cyclooxygenase-2 (COX-2) and Interleukin-1 β (IL-1β) in the NF-кB pathway in mice with colitis, significantly alleviating the pathological condition of the mice.

The core of traditional Chinese medicine fermentation lies in microbial transformation [21], a process which relies on the abundant enzyme system in microorganisms to catalyze reactions in raw materials [22]. Biphasic fermentation technology, an evolution of solid fermentation, combines Chinese medicinal materials with edible and medicinal fungi under specific conditions. This process provides nutrients for bacteria and simultaneously alters the Chinese medicinal base material through the fungal enzyme system, resulting in complex bacteria with specific pharmacological activities, such as anti-inflammatory and antitumor activity. Li et al. [23] utilized biphasic fermentation technology to biotransform *Marsdenia tenacissima* (MT) with *Ganoderma lucidum*, identifying polysaccharides, saponins, organic acid alkaloids, and flavonoids of *Ganoderma lucidum* co-fermentation of *MT* by non-targeted metabolomics analysis and stoichiometric analysis. The study revealed a significant increase in amino acids and organic acids among the 249 annotated metabolites. This process not only improved the bitter taste of the original medicine but also enhanced its anti-inflammatory, antibacterial, and anticancer activities, displaying an inhibitory effect on lung cancer.

Recent studies have focused on applying fermentation technology to develop the food and medicinal value of agricultural by-product waste [24]. However, research on the utilization of *Trapa bispinosa* shells as a by-product of agricultural production remains limited. In this study, we employed a response surface experiment to optimize the fermentation conditions of TBs for *Ganoderma sinense*. Polysaccharides and polyphenols generated through fermentation were characterized using various analytical techniques. Subsequently, the immune-boosting and antioxidant properties of the fermented extract were assessed in mouse macrophage RAW264.7 cells.

## 2. Results

### 2.1. Single-Factor Fermentation Optimization Results

Single-factor experiments were conducted to assess the influence of fermentation time, temperature, and humidity on TBs. The water extract yield and DPPH free radical scavenging rate were measured as indicators. The results indicated that the highest water extract yield and DPPH radical scavenging rate were achieved at 2 days of fermentation, 15 °C, and 75% humidity (Figure 1). Notably, significant differences were observed between different gradients within the same single-factor group. To enhance the precision of subsequent response surface experiments and achieve optimal outcomes, the following parameters were selected: fermentation time options of 1.5 days, 2 days, and 2.5 days; fermentation temperature options of 12.5 °C, 15 °C, and 17.5 °C; and fermentation humidity options of 70%, 75%, and 80%.

### 2.2. Response Surface Fermentation Optimization Results

The experimental results, with the water extract yield as the response value, are presented in Table 1. To enhance clarity, a 3D space diagram was generated for analysis. The steepness of the slope in the 3D image of the response surface indicates the significance of the condition’s effect on the change in the response value, i.e., the yield of water extract. Conversely, a less steep slope signifies a smaller effect of that factor on the water extract extraction rate. As shown in Figure 2, among the pairwise interactions of various factors affecting the yield of total flavonoids, the interaction between fermentation temperature (X_2_) and fermentation humidity (X_3_) was significant. In contrast, the interactions between fermentation time (X_1_) and fermentation temperature (X_2_), as well as fermentation time (X_1_) and fermentation humidity (X_3_), had no significant effect on the yield of flavonoids from *Trapa bispinosa* shells, aligning with the results of variance analysis.

The regression equation derived from regression analysis using the Design Expert 11 software is as follows: Y = −95.0159 + 12.3334X_1_ + 1.8771X_2_ + 1.9424X_3_ − 0.0206X_1_X_2_ + 0.0106X_1_X_3_ − 0.0078X_2_X_3_ − 3.3119X_1_^2^ − 0.0441X_2_^2^ − 0.0121X_3_^2^, with an R^2^ value of 0.9958. As indicated in Appendix A, the model has high significance with *p* < 0.0001. The lack-of-fit term (*p* > 0.05) indicated a well-fitted equation.

The optimized process parameters for GS fermentation of TBs were determined using the Design Expert software. The optimal extraction process comprised a fermentation duration of 1.93 days, a temperature of 14.1 °C, and a humidity of 76.53%, resulting in a corresponding water extract yield of 4.52%. For practical purposes, the optimal conditions were adjusted to a fermentation time of 2 days, a fermentation temperature of 14 °C, and a fermentation humidity of 77%. The experiment was repeated three times to validate the process conditions, yielding a water extract yield of TBs at 4.36 ± 0.02%. The actual value closely matched the predicted value of the model, showing no significant difference, indicating the model’s feasibility. Therefore, in subsequent experiments, the chosen fermentation conditions were a fermentation time of 2 days, a temperature of 14 °C, and a fermentation humidity of 77%. Under these conditions, the DPPH radical scavenging rate of the aqueous extract obtained was 82.36 ± 0.06%.

### 2.3. Changes in the Activity of Purified Polysaccharides and Alcoholic Extracts before and after Fermentation

The effect of various concentrations of polysaccharides and alcoholic extracts on the viability of RAW264.7 cells was assessed using the CCK-8 method. The results are displayed in Figure 3. When the concentration of P1 and P2 exceeded 150 μg/mL, the cells exhibited damage and apoptosis. Similarly, when the concentration of AE1 and AE2 surpassed 150 μg/mL, the cells also displayed damage and apoptosis. Therefore, three concentrations of 50 μg/mL, 100 μg/mL, and 150 μg/mL were selected as the processing quantities in the subsequent P1, P2, AE1, and AE2 experiments. In addition, the effects of various concentrations of LPS and H_2_O_2_ on the viability of RAW264.7 cells were assessed using the CCK-8 method. The NO secretion of RAW264.7 cells treated with different concentrations of LPS was measured using the Griess method. The results are shown in Figure 3. When the concentration of LPS exceeded 1 μg/mL, the cells exhibited damage and apoptosis, while the concentration was 0.5 μg/mL. The secretion of nitric oxide (NO) reached the highest level, so the concentration of 0.5 μg/mL was set as the concentration of the lipopolysaccharide (LPS) positive control. When the concentration of H_2_O_2_ reached 20 mmol/mL, the cell survival rate was less than 50%; therefore, 20 mmol/mL was chosen as the H_2_O_2_ stimulation concentration.

Figure 4 illustrates that, following treatment with P1 and P2, compared to the blank control group, the secretion levels of NO and iNOS significantly increased in a concentration-dependent manner. Notably, the content of NO and iNOS stimulated by P2 was markedly higher than that in the P1 group at different concentrations. Furthermore, in comparison to the blank control group, the secretion levels of TNF-α, IL-2, and IL-10 exhibited a concentration-dependent increase, with P2 inducing significantly higher levels than P1 at various concentrations.

As shown in Figure 5, following treatment with AE1 and AE2, T-AOC and SOD showed a significant increase, while Malondialdehyde (MDA) secretion exhibited a significant decrease. MDA is one of the final products of polyunsaturated fatty acid peroxidation by H_2_O_2_ in the cell.

To investigate the causes of these functional changes, the structure of the polysaccharides and the components of the alcoholic extract were analyzed.

### 2.4. Changes in the mRNA Expression Level on the NF-κB Pathway and Nrf2/Keap1-ARE Pathway

As shown in Figure 4E, upon treatment with LPS at a concentration of 0.5 μg/mL for 24 h, the gene expression of *p52* and *ikkα* was significantly downregulated, while *p65* was significantly increased. However, the gene expression of *p50* and *p100* did not show significant changes under this treatment condition. In comparison to the blank group, the expression levels of the *p50*, *p52*, *p65*, and *p100* genes were significantly upregulated after treatment with 150 µg/mL of both P1 and P2 for 24 h (*p* < 0.05). Conversely, the expression level of the *ikkα* gene was significantly decreased (*p* < 0.05), and the effect of P2 was notably higher than P1 (*p* < 0.05). These findings suggest that both P1 and P2 could promote immune regulation through the NF-кB pathway, with P2 exhibiting a significantly better effect than P1.

As shown in Figure 5D, after treatment with 150 µg/mL L-Ascorbic acid and P2, the expression of the *keap1* gene was significantly decreased (*p* < 0.05), and the expression of the *gpx2*, *gpx5*, *gclc*, and *gclm* genes was significantly increased (*p* < 0.05). Among these, AE2 showed no significant difference with L-Ascorbic acid in the upregulation of *gpx5* gene expression, and the others were less potent than L-Ascorbic acid. However, AE1 induced a significant upregulatory effect on the expression of the *gpx2*, *gpx5*, and *gclc* genes, although it was weaker than AE2. The results indicate that both AE1 and AE2 could exert antioxidant effects through the Nrf2/Keap1-ARE pathway, with the effect of AE2 being significantly better than that of AE1.

### 2.5. Analysis of the Structural Characteristics of Polysaccharides after Fermentation

#### 2.5.1. The Molecular Weight of Polysaccharides after Fermentation

The absolute molecular masses of P1 and P2 were determined using the SEC-MALL-RI system, as depicted in Figure 6. The chromatograms of P2 displayed multiple peaks, indicating that P2 comprised multiple fractions with different molecular mass intervals. The molecular weight of the polysaccharides was directly proportional to the intensity of the light scattering peak, suggesting a broad distribution of polysaccharides with varying molecular weights, particularly in the low molecular weight region. In terms of differential detection peaks, P1 and P2 exhibited different degrees of shoulder peaks. P1’s peak was relatively scattered, while P2 featured a peak in the high molecular mass region (10–15 min). Consequently, P1 and P2 were not characterized by a single molecular mass level.

The content distribution of fractions with different weight-average molecular masses in P1 was relatively homogeneous (Table 2). In contrast, the weight-average molecular mass of the main fraction in P2 was 2.019 × 10^5^ ± 1.112 u. The average molecular weight of P1 and P2 was 6.004 × 10^6^ u and 7.812 × 10^4^ u, respectively, indicating a decrease in molecular weight after fermentation.

#### 2.5.2. The Morphology of Polysaccharides after Fermentation

Scanning electron microscopy (SEM) images of P1 and P2 are presented in Figure 7. In terms of appearance, P1 exhibited regular, mostly granular shapes arranged closely and orderly. In contrast, P2 displayed different characteristics, appearing crystalline, blocky, unevenly distributed, and more closely arranged. Regarding particle uniformity, P1 had relatively uniform particles, while P2 exhibited varying particle sizes.

#### 2.5.3. The Characteristic Functional Group Structure of Polysaccharides

Figure 8 illustrates the characteristic spectra of P1 and P2. Notable features included polysaccharide characteristic peaks such as hydroxyl O-H vibration peak (3420 cm⁻^1^), CH_2_ (1457 cm⁻^1^), and skeleton vibration (600–400 cm⁻^1^). There were characteristic α-glycosidic bond peaks (1024 cm⁻^1^ and 840 cm⁻^1^) and a peak characteristic of β-glucan (1081 cm⁻^1^). Comparing P1 and P2, changes included strengthened vibration intensity of CH_2_ in the polysaccharide skeleton, a shift in the peak of amide I towards a lower wavenumber, increased vibration intensity of the hydroxyl group, and other specific changes in corresponding structures outlined in Appendix A.

#### 2.5.4. The Triple Helix Structure of Polysaccharides

In Figure 9, the maximum absorption wavelengths of polysaccharides before and after fermentation were 484 nm and 487 nm, respectively, falling between 400 nm and 650 nm. Compared with the blank group (475 nm), the maximum absorption wavelengths of polysaccharides exhibited a red shift by 9 nm and 12 nm, respectively. These results indicated that the polysaccharides before and after fermentation exhibited a certain triple helix structure, with the fermented polysaccharides showing a more pronounced structure.

### 2.6. Analysis of the Components of Alcoholic Extracts after Fermentation

Utilizing the UPLC-QTOF-MS/MS method, a total of 960 substances were detected, with 66 substances identified exhibiting a relative percentage higher than 0.05%. These substances encompassed polyphenols, flavonoids, alkaloids, triterpenoids, ganoderic acid components, and new or disappeared components after fermentation. The alterations in each substance are detailed in Appendix A. Specifically, there were 58 compounds in the AE1 part and 63 compounds in the AE2 part. The top 20 relative compounds were considered principal components. A comparison between AE1 and AE2 revealed a reduction in polyphenolic compounds, such as [6]-gingerol, (-)-gallocatechin, and cynaroside, while flavonoids showed the addition of myricetin and maltol and a reduction in alkaloids like DL-tyrosine. Triterpenoids in AE2 showed the addition of ganoderol A, linderalactone, and lamiide, with a reduction in artemisinin.

## 3. Materials and Methods

### 3.1. Experiment Materials

The experiment materials encompassed a range of chemicals and biological substances essential for this study. These included the 1,1-Diphenyl-2-trinitrophenylhydrazine (DPPH) standard, formic acid, methanol, acetonitrile, and chromatographic-grade potassium bromide (KBr) from Sigma-Aldrich, Darmstadt, German. Additionally, supplies such as DMEM high-glucose culture, phosphate buffer (PBS), fetal bovine serum (FBS), penicillin and streptomycin solution, cell viability assay kit (CCK-8), dimethyl sulfoxide (DMSO), and lipopolysaccharide (LPS) isolated from Escherichia coli 0111:B4 were obtained from Sigma-Aldrich. Specialized assay kits for Nitric Oxide (NO), Nitric Oxide Synthase Activity (NOS), total antioxidant capacity, Superoxide Dismutase (SOD), and Malondialdehyde (MDA) were sourced from Nanjing Jiancheng Biological Engineering Institute, Nanjing, China. ELISA kits for TNF-α, IL-2, and IL-10 were acquired from Merck Millipore. RNAiso plus, reverse transcription kit, and fluorescence quantitative kit were procured from TAKARA, Japan, while primer sequences were obtained from Shanghai Sangon Biological Co., Ltd., Shanghai, China.

The biological materials utilized in the experiment included *Trapa bispinosa* with a green shell, purchased from Honghu, China. Additionally, *Ganoderma sinense* bio-57607 and mouse macrophage (RAW264.7) were obtained from the Chinese Typical Culture Collection Center in Wuhan, China.

### 3.2. Optimization of the Fermentation Process

The single-factor experiments were designed with three key factors: fermentation time, temperature, and humidity. For fermentation time, durations of 0, 1, 2, 3, 4, and 5 days were chosen, while the other factors were held constant at 25 °C and 75% humidity. Fermentation temperatures of 10, 15, 20, 25, and 30 °C were selected, and the fermentation was conducted under 75% humidity for 3 days. Humidity levels of 55, 65, 75, 85, and 95% (relative humidity) were chosen, and the fermentation was carried out at 25 °C for 3 days. Following fermentation, the fungi were desiccated and pulverized under dark conditions. Distilled water was added at a ratio of 1:100 for extraction. The resulting liquid extract underwent filtration and was subsequently compressed under reduced pressure at 60 °C. The compressed extract was then subjected to vacuum freeze-drying to obtain the powder. The water extraction rate and the DPPH free radical scavenging activity, indicative of antioxidant potential, were determined. From each of the three single-factor experiments, three optimal factors were selected as variable factors. Subsequently, a response surface experiment with three factors and three levels was designed (Table 3). The formulation of the response surface experiment is detailed in Appendix A, based on the results obtained from the single-factor experiments.

### 3.3. Preparation of Purified Polysaccharides and Alcoholic Extracts from TBs

The products, both before and after fermentation, underwent extraction with 90 °C hot water at a solid–liquid ratio of 1:10 for 3 h, with the process repeated once. The resulting extracts were combined. Centrifugation at 10,000 r/min (9168× *g*) for 10 min was employed to harvest the supernatant. Utilizing the Sevag method with reference to Niu FL et al.’s method [25], the supernatant underwent treatment with a 4:1 chloroform-n-butanol solution equivalent to 1/4 volume. Manual shaking for 10 min was followed by 20 min standing period before collecting the aqueous phase through a separating funnel. This process was iterated three times. The resulting aqueous phase was combined with three times its volume of absolute ethanol, mixed at 4 °C for 12 h, and then centrifuged at 4500 r/min (1856× *g*) for 10 min. The precipitate was removed, and the crude extracted polysaccharide was obtained through freeze-drying. Next, 100 mg of the crude extracted polysaccharide was dissolved in 20 mL of ddH_2_O, filtered through a 0.45 μm microporous filter membrane, and slowly dropped onto a DEAE-cellulose 52-anion exchange column (26 mm×40 mm). Discharging at the lower end of the exchange column, elution with ultra-pure water at a rate of 1 mL/min took place. Polysaccharide content in each tube was determined using the phenol-sulfuric acid method. The elution curve was plotted with the number of collected tubes as the abscissa and the absorbance of each tube measured at 490 nm as the ordinate. Peak segments with more fractions were chosen for sample collection, and the purified polysaccharide samples were obtained through vacuum freeze-drying. The purified polysaccharide samples before fermentation were denoted as P1 and after fermentation as P2.

The entire product before and after fermentation was extracted using ethanol. An appropriate amount of samples was immersed in 70% ethanol with a solid–liquid ratio of 1:100. After stable magnetic stirring at 60 °C for 30 min, ultrasonic extraction at 50 °C for 45 min, and constant temperature magnetic stirring at 60 °C for an additional 30 min, the obtained extract was concentrated under reduced pressure at 60 °C post filtration and vacuum freeze-drying, yielding the alcoholic extract samples. The alcoholic extract before fermentation was named AE1, and the alcoholic extract after fermentation was named AE2. These groups of purified polysaccharides and alcoholic extracts were then prepared for subsequent experiments.

### 3.4. Analysis of Changes in Immunomodulatory and Antioxidant Activities during TBs’ Fermentation

The impact of varying concentrations of polysaccharide extracts on RAW264.7 cell viability was assessed using the CCK-8 assay. The three highest concentrations of purified polysaccharides, exhibiting minimal cytotoxicity, were chosen for subsequent experiments. The influence of the positive control LPS on RAW264.7 activity was also determined using the CCK-8 assay. The Griess method was employed to measure the secretion of NO in RAW264.7 cells treated with the positive control LPS, and a standard curve was generated to identify the LPS concentration with the highest NO secretion that did not adversely affect RAW264.7 cell activity. This concentration was deemed safe for subsequent experiments. Based on experimental results, concentrations of 50, 100, and 150 µg/mL were selected for treatment with purified polysaccharides (P1 and P2), and the LPS positive control concentration was set to 0.5 µg/mL to establish an LPS positive control model for subsequent experiments. The secretion of NO, iNOS, TNF-α, IL-2, and IL-10 in RAW264.7 cells under the influence of different concentrations of P1 and P2 was detected. The RAW264.7 cells’ secretion of the blank group and the positive group were compared to determine the changes in immunomodulatory activity before and after fermentation.

Similarly, the effect of different concentrations of alcoholic extracts on RAW264.7 cell viability was determined using the CCK-8 assay, along with the positive control H_2_O_2_. The three highest concentrations of alcoholic extracts, which exhibited no cytotoxic effects, were selected for subsequent experiments. An oxidative stress model of RAW264.7 cells was constructed using H_2_O_2_, with a concentration resulting in a survival rate of less than 50%. Based on the experimental results, concentrations of 50, 100, and 150 µg/mL were chosen for the extract treatment in subsequent experiments. The concentration of 20 mmol/mL was selected for H_2_O_2_ stimulation to construct the oxidative stress model. The levels of total antioxidant capacity (T-AOC), Superoxide Dismutase (SOD), and Malondialdehyde (MDA) in RAW264.7 cells under the influence of different concentrations of AE1 and AE2 were detected. The RAW264.7 cells’ secretion of the blank group and the positive group were compared to determine the changes in immunomodulatory activity before and after fermentation. The RAW264.7 cells’ secretion of the blank group and the LPS positive control group were compared to determine the changes in antioxidant activity before and after fermentation.

### 3.5. Polysaccharides Structure Characterization of TBs Fermented by GS

The molecular weight of polysaccharides was determined through high-performance size-exclusion chromatography. To prepare the sample solution, 5 mg of each P1 and P2 was dissolved in 2 mL of 0.2 mol/L ammonium acetate solution, centrifuged at 12,000 r/min (13,201× *g*) for 5 min, and filtered through a microporous filter membrane with a pore size of 0.22 μm. Dextran standards with varying molecular weights (5 mg) were accurately weighed and dissolved in 2 mL of a 0.2 mol/L ammonium acetate solution. After vigorous shaking, the solution was centrifuged at a speed of 13,000 r/min (15,493× *g*) for 5 min, followed by filtration through a microporous filter membrane with a pore size of 0.22 μm to prepare the standard solution for future use. High-performance molecular-exclusion chromatography was conducted using an Agilent PL Aquagel-OH gel column with 0.2 mol/L ammonium acetate as the mobile phase. A sample volume of 20 μL was injected at a flow rate of 0.7 mL/min and maintained at a temperature of 30 °C.

The morphology of the polysaccharides was observed using scanning electron microscopy. Electron microscope samples were prepared by ion sputter coating fully dried P1 and P2 gold-sprayed coatings. The acceleration voltage was adjusted to 200–30,000 V, and the magnification ranged from 10^4^ to 10^5^ times. The surface morphology of P1 and P2 particles was examined.

The functional group structure of polysaccharides was analyzed using Fourier transform infrared (FT-IR) spectroscopy. Pure KBr powder slides were used to adjust the baseline. P1 and P2 were mixed with appropriate amounts of KBr powder, ground evenly in an agate mortar to make thin slices, and then subjected to infrared scanning at 4 Hz and 32 over a wavelength range of 400–4000 cm^−1^.

The Congo red method with reference to Bao X F et al.’s method [26] was employed to resolve the triple helix structure of polysaccharides. Specifically, 1 mL of a 2 mg/mL sample solution was mixed with 3 mL of NaOH solution (0.2 moL/L), followed by the addition of 1.5 mL of a 0.1 mmol/L Congo red solution and enough deionized water to bring the total volume to 5 mL. The resulting mixture was thoroughly mixed and allowed to stand at room temperature for one hour. Deionized water was utilized as the blank control group, and a full-spectrum UV scanner was employed to measure the maximum absorption wavelength of the reaction solution within 400–800 nm. By plotting the trend chart with wavelength as abscissa and corresponding absorption value as ordinate, a comparison with the blank control was conducted to analyze P1 and P2’s triple helix structure.

### 3.6. Component Analysis of Alcoholic Extract of TBs Fermented by GS

Ultra-high performance liquid chromatography–quadrupole time-of-flight tandem mass spectrometry (UPLC-QTOF-MS/MS) technology was employed to analyze and determine the composition of AE1 and AE2 samples. Mass spectrum data collection and quantitative analysis were conducted on target compounds using SCIEX Analyst Workstation Software (Version 1.6.3). The original mass spectra were converted to “txt” format, and a self-written R package, combined with a self-built database, was used to perform peak extraction, annotation, and other analyses.

The chromatography–mass spectrometry conditions were as follows: ultra-performance liquid chromatography with a Waters Acquity UPLC BEH C18 (1.7 μm, 2.1 × 100 mm) column. The mobile phase A consisted of 0.1% formic acid in water, while B was acetonitrile. The injection volume was 2 μL, and the flow rate was set at 0.4 mL/min. The column temperature was maintained at 40 °C, and the detection wavelength was set at 425 nm. The gradient elution conditions were as follows: 0 min, 2% B; 0.5 min, 2% B; 10 min, 50% B; 11 min, 5% B; 13 min, 5% B; 13.1 min, 98% B; 15 min, 98% B. Mass spectrometry was performed in multiple reaction monitoring (MRM) mode on a SCIEX 6500 QTRAP+ triple quadrupole MS spectrometer equipped with IonDrive Turbo V ESI ion source. Ion source parameters included electrospray ion source, positive ion mode acquisition, ion spray voltage of −4500 V/+5500, ESI ion source temperature of 500 °C, curtain air at 35 psi, and drying gas and auxiliary drying gas at 60 psi with a flow rate of 8.0 L/min and a temperature of 400 °C. The collision dissociation energy was 10 eV in the first stage of scanning and 35 eV in the second stage of scanning, with a fluctuation range of ±10 eV. The uncluster voltage was set at 100 V. The full scan mass number range was *m*/*z* 100–1500 for the first stage of scanning and *m*/*z* 50–1500 for the second stage of scanning.

### 3.7. Analysis of the mRNA Expression Levels on the NF-κB Pathway and Nrf2/Keap1-ARE Pathway

To investigate the changes in the expression levels of NF-κB pathway-related genes, four experimental groups were established: the control group, the 0.5 μg/mL LPS positive group, the 150 μg/mL P1 intervention group, and the 150 μg/mL P2 intervention group, each consisting of six replicate wells. Cells were seeded into 6-well plates with 2 mL of a 5 × 10^5^ cells/mL per well and cultured for 24 h in an incubator at 37 °C.

Similarly, to investigate the changes in the expression levels of Nrf2/Keap1-ARE pathway-related genes, four experimental groups were analyzed: the control group, the 150 µg/mL L-Ascorbic acid (VC) positive group, the 150 µg/mL AE1 intervention group, and the 150 µg/mL AE2 intervention group, each with six replicate wells. Cells were seeded into 6-well plates with 2 mL of a 5 × 10^5^ cells/mL per well and cultured for 24 h in an incubator at 37 °C.

In both experiments, the experimental groups were treated with a medium containing different concentrations of the corresponding extract, and the blank group was treated with a normal medium. The culture was continued for 24 h. Total RNA was extracted using the RNAiso plus kit, and the RNA concentration (ng/μL) was determined by an ultra-micro UV-visible spectrophotometer. After purification of the extract solution, the first-strand cDNA was synthesized using a Takara RT-PCR kit. Primer sequences for NF-κB pathway-related genes and Nrf2/Keap1-ARE pathway-related genes were utilized (Appendix A). Real-time PCR was performed with the following cycling conditions: 95 °C for 3 min, 95 °C for 15 s, 53 °C for 30 s, 72 °C for 20 s, and 40 cycles. The housekeeping gene used mouse ACTB gene (primer bought in Sangon, Shanghai, id B662302), and the gene expression levels were analyzed by 2^−ΔΔCt^ method. The mRNA expression levels of NF-κB pathway-related genes and Nrf2/Keap1-ARE pathway-related genes were determined and compared based on the amplification curve data.

### 3.8. Statistical Analysis

All data were expressed as mean ± standard deviation (SD). Statistical analysis and graphs were generated using GraphPad prism 8.0.2 software. Any *p* values less than 0.05 were considered significant.

## 4. Discussion

In the present study, under optimized fermentation conditions, various aspects of TBs polysaccharides fermented by GS were examined, encompassing molecular weight, apparent morphology, functional group structure, and triple helix structure. Polysaccharides featuring a triple helix structure were examined for their interaction with acid Congo red, inducing a redshift in UV absorption wavelength. Conversely, the presence of NaOH disrupted the complexation structure, leading to a decrease in the maximum absorption wavelength. This methodology facilitated the determination of the polysaccharides’ triple helix structure. These findings revealed a significant decrease in the molecular weight of the fermented polysaccharides. Scanning electron microscopy images demonstrated alterations in the shape, arrangement, and particle uniformity of the fermented polysaccharides. Fourier transform infrared spectroscopy and acid Congo red staining ultraviolet scanning hinted at changes potentially arising from shifts in functional groups and an augmented triple helix structure within the polysaccharides.

It is now understood that NO serves as a crucial immunomodulatory factor involved in the intricacies of immune regulation. Its multifaceted roles include acting as a bactericidal agent, stimulating immune activity, and serving as a pivotal signal for monocytes and macrophages to exert immune effects like antibacterial and antitumor activities [27]. It is well established that inducible nitric oxide synthase (iNOS) plays a role in promoting NO production and aiding in the initiation of immune activity in macrophage RAW264.7, particularly at the onset of inflammation [28]. Tumor necrosis factor-α (TNF-α), originating from diverse cell types, regulates cell growth, orchestrates immune system functions, and sparks immune activity in the initial stages of inflammation [29]. Interleukin-2 (IL-2), a growth factor within the immune system, contributes to regulating the activity of white blood cells, participating in processes such as antibody response, tumor surveillance, and hematopoiesis [30,31,32]. Interleukin-10 (IL-10), an inhibitor of cytokine synthesis (CSIF), is a versatile cytokine with immunomodulatory functions across various cell types [33]. NF-кB, a protein complex formed through the dimerization of Rel A/P65, P52, C-Rel, P50, and its precursor P105 [34,35], is integral to the immune response across a spectrum of diseases, primarily associated with tissue damage, stress, body defense response, cell apoptosis, and differentiation. Additionally, it plays a role in information transmission during the process of tumor cell growth and inhibition [36]. In this study, to determine the changes in the immunomodulatory effects of the extract before and after fermentation, NO, iNOS, TNF-α, IL-2, and IL-10 were employed as evaluation criteria. The outcomes from RAW264.7 cells stimulated by TBs before and after GS fermentation indicated a significant increase in the inflammatory response triggered by the fermented polysaccharides compared to before fermentation. This suggests that GS fermentation could enhance the pro-inflammatory activity of polysaccharides extracted from TBs. These results further revealed a significant upregulation in the expression of NF-кB-related genes in RAW264.7 cells stimulated by polysaccharides after fermentation compared to before fermentation, suggesting that GS fermentation could promote the pro-inflammatory activity of RAW264.7 cells, potentially achieved by stimulating the expression of NF-кB-related genes.

Total antioxidant capacity (T-AOC) represents the collective ability of antioxidant substances in the body to combat oxidative damage [37]. Superoxide Dismutase, known for its specialized physiological activity, plays a crucial role in scavenging free radicals within organisms. Malondialdehyde stands as a key product of lipid peroxidation in cell membranes, contributing to aggravated cell membrane damage and expediting the cellular aging process [38]. NF-E2-related factor 2 (Nrf2), a pivotal factor in the cellular oxidative stress response, is intricately regulated by Keap1 and interacts with the antioxidant response element (ARE), playing a vital role in maintaining cellular redox homeostasis [38,39]. In this study, T-AOC, SOD, and MDA served as evaluation criteria. The results obtained from RAW264.7 cells stimulated by the alcoholic extract of TBs before and after GS fermentation demonstrated a significant enhancement in the antioxidant capacity of the alcoholic extract after fermentation, surpassing the improvement observed in the alcoholic extract before fermentation. Additionally, the findings revealed that the alcoholic extract stimulation after fermentation markedly increased the expression of Nrf2-Keap1-ARE pathway genes in RAW264.7 cells compared to before fermentation. This suggests that GS fermentation could enhance the antioxidant activity of RAW264.7 cells, potentially achieved by upregulating the expression of Nrf2-Keap1-ARE pathway genes.

Sun H et al. [40] showed that polysaccharides extracted from fermented Yupingfeng are significantly better than those from unfermented products in upregulating the secretion of NO, iNOS, IL-6, TNF-α, and INF-γ. This suggests that fermentation enhances the immunomodulating activity. Liang Y et al.’s [41] research showed that a polysaccharide from plant fermentation extracts could significantly enhance the phagocytic activity and secretion of NO, iNOS, TNF-α, IL-6, and IL-10 of RAW264.7. Xie L et al.’s [42] study showed that the polysaccharides from liquid fermentation of Monascus purpureus 40,269 could activate RAW264.7 cells, promoting the secretion of reactive oxygen species, NO, IL-1, TNF-α, and IL-6. This demonstrates that these polysaccharides could be used as a functional food for immunological reagents. Tang H et al.’s [43] research showed that polysaccharides from fermented mycelia of Coriolus versicolor could alleviate the symptoms of NAFLD in an HFD-induced mice model. Hu W et al. [44] showed that the purified polysaccharides from Hericium erinaceus fermented mycelium could improve the cognitive behavior of mice. The anti-Alzheimer’s disease (AD) property was achieved by regulating calcium homeostasis mediated by oxidative stress.

In conclusion, the fermentation of TBs by GS catalyzes the transformation reaction of TBs components through GS’s abundant enzyme system. This leads to a decrease in the molecular weight of TBs polysaccharides, an increase in the content of the triple helix structure, and the transformation of characteristic functional groups in the polysaccharides. These changes may impact the absorption and utilization of the polysaccharides, thereby affecting their medicinal value. Furthermore, after fermentation, there is an increase in the content of compounds with significant active functions in the alcohol-extracted small molecule fraction, which may also contribute to the observed changes in the biological activity.

## Figures and Tables

**Figure 1 molecules-29-01238-f001:**
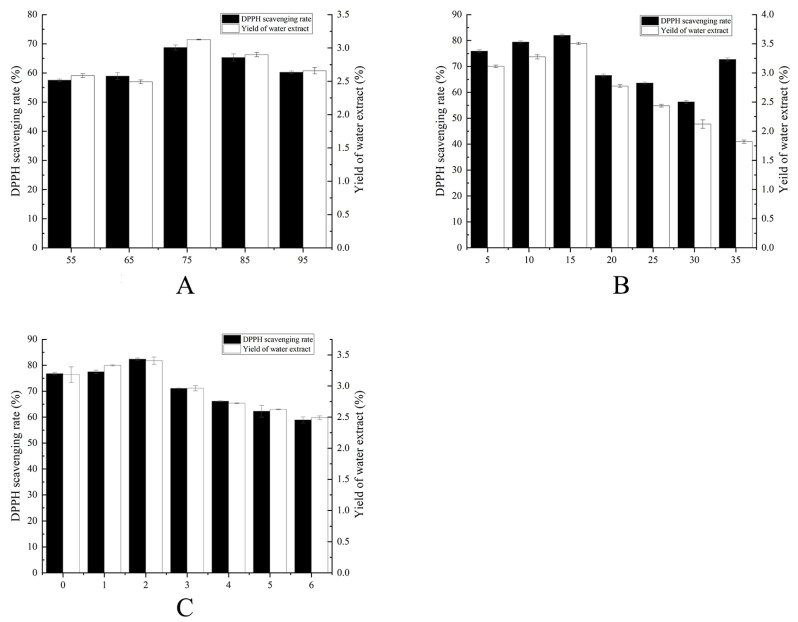
The single-factor experiment plots show the rate of DPPH radical scavenging and yield of water extract: (**A**) Fermentation humidity (%); (**B**) Fermentation temperature (°C); (**C**) Fermentation times (d).

**Figure 2 molecules-29-01238-f002:**
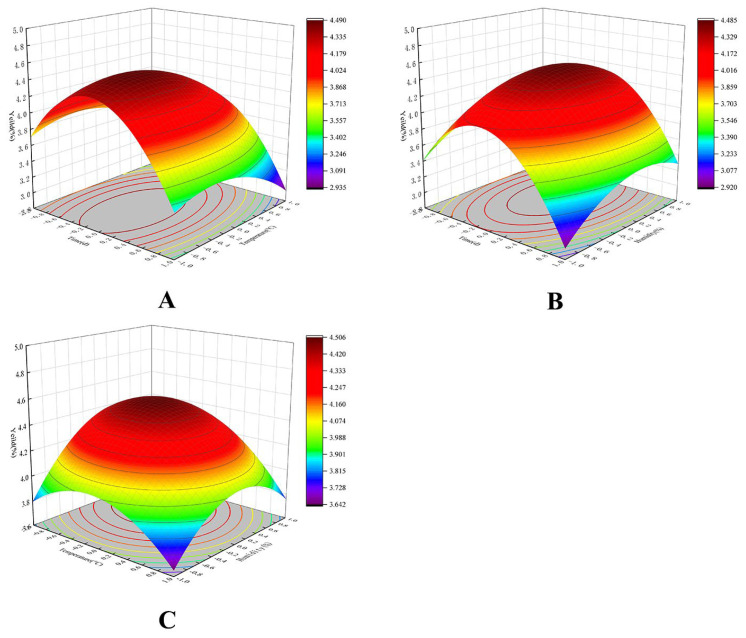
Response surface plots showing effects of variables on the extraction yield of water extract: (**A**) Interaction of Temperature and Time; (**B**) Interaction of Humidity and Time; (**C**) Interaction of Humidity and Temperature.

**Figure 3 molecules-29-01238-f003:**
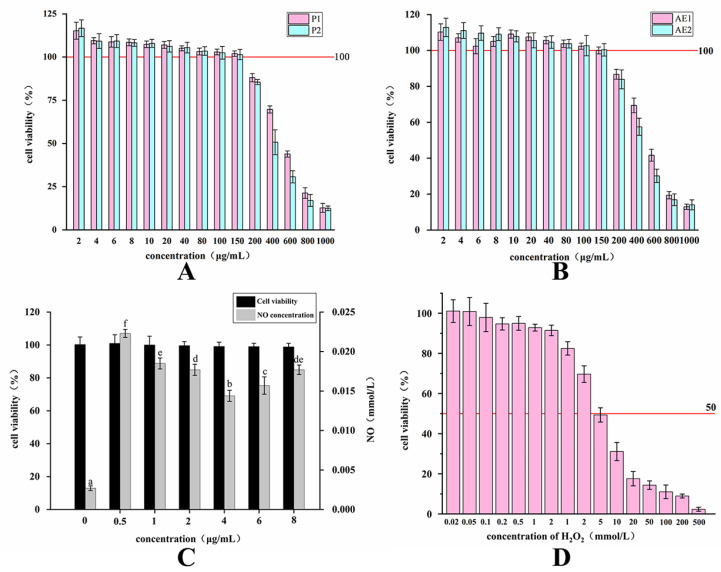
The cytotoxicity models of RAW264.7 cells: (**A**) cytotoxicity of P1, P2 treatment; (**B**) cytotoxicity of AE1, AE2 treatment; (**C**) cytotoxicity and NO secretion of LPS treatment. The values in the figure are labeled with superscripts a-f in order from lowest to highest. Values with different superscripts are significantly different (*p* < 0.05). (**D**) cytotoxicity of H_2_O_2_ treatment.

**Figure 4 molecules-29-01238-f004:**
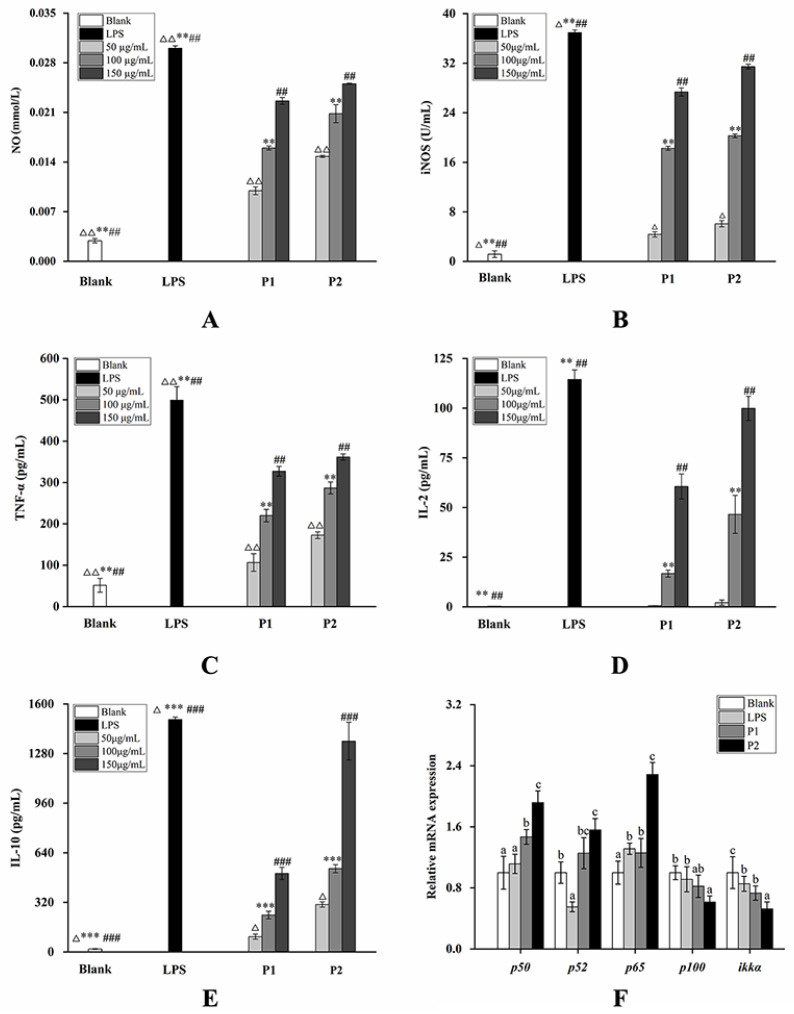
The effect on inflammatory factors of P1 and P2: (**A**) secretion of NO; (**B**) secretion of iNOS; (**C**) secretion of TNF-α; (**D**) secretion of IL-2; (**E**) secretion of IL-10; (**F**)expression of genes related to NF-κB pathway. The values in the figure are labeled with superscripts a-c in order from lowest to highest. Values with different superscripts are significantly different (*p* < 0.05). Notes: 50 µg/mL P1, P2 compared with LPS treatment group and blank control group, ^Δ^
*p* < 0.05, ^ΔΔ^
*p* < 0.01; 100 µg/mL P1, P2 compared with LPS treatment group and blank control group, ** *p* < 0.01, *** *p* < 0.001; 150 µg/mL P1, P2 compared with LPS treatment group and blank control group, ^##^
*p* < 0.01, ^###^
*p* < 0.001.

**Figure 5 molecules-29-01238-f005:**
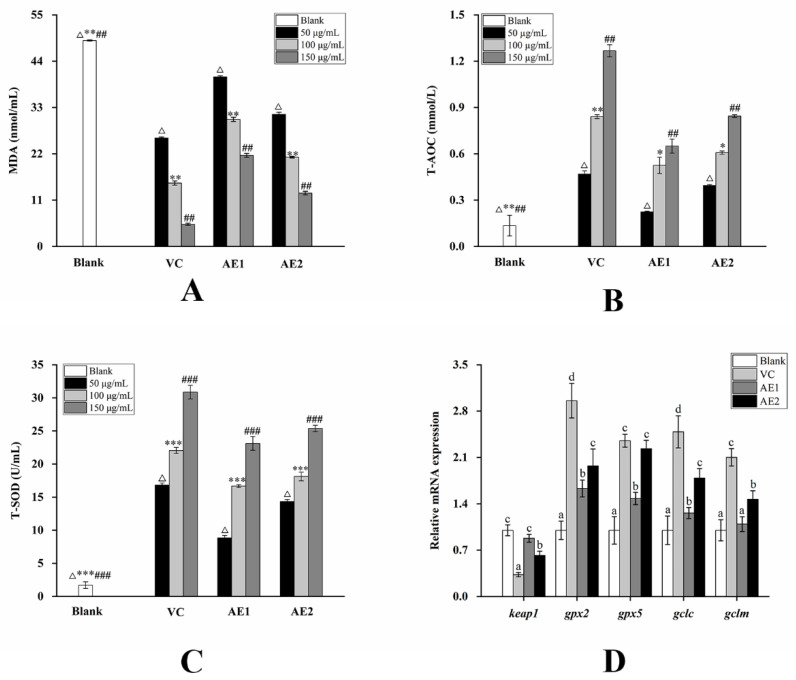
The effects on antioxidant activity of AE1 and AE2: (**A**) secretion of MDA; (**B**) secretion of T-AOC; (**C**) secretion of T-SOD; (**D**) expression of genes related to Nrf2/Keap1-ARE pathway. The values in the figure are labeled with superscripts a-c in order from lowest to highest. Values with different superscripts are significantly different (*p* < 0.05). Note: 50 µg/mL L-Ascorbic acid (VC), AE1, AE2 treatment group compared with blank control group, ^Δ^
*p* < 0.05; 100 µg/mL L-Ascorbic acid (VC), AE1, AE2 treatment group compared with blank control group, * *p* < 0.05, ** *p* < 0.01, *** *p* < 0.001; 150 µg/mL L-Ascorbic acid (VC), AE1, AE2 treatment group compared with blank control group, ^##^
*p* < 0.01, ^###^
*p* < 0.001.

**Figure 6 molecules-29-01238-f006:**
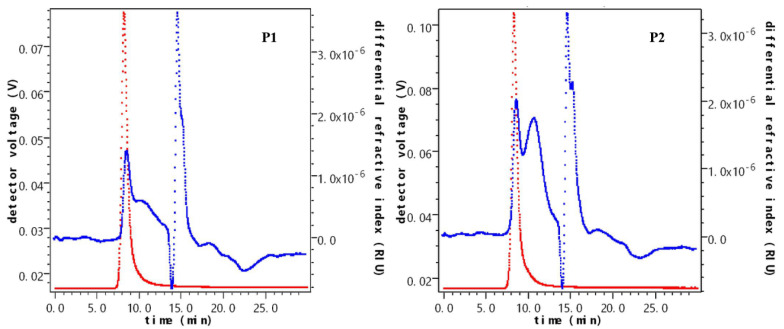
SEC−MALL−RI chromatograms of (**P1**) and (**P2**).

**Figure 7 molecules-29-01238-f007:**
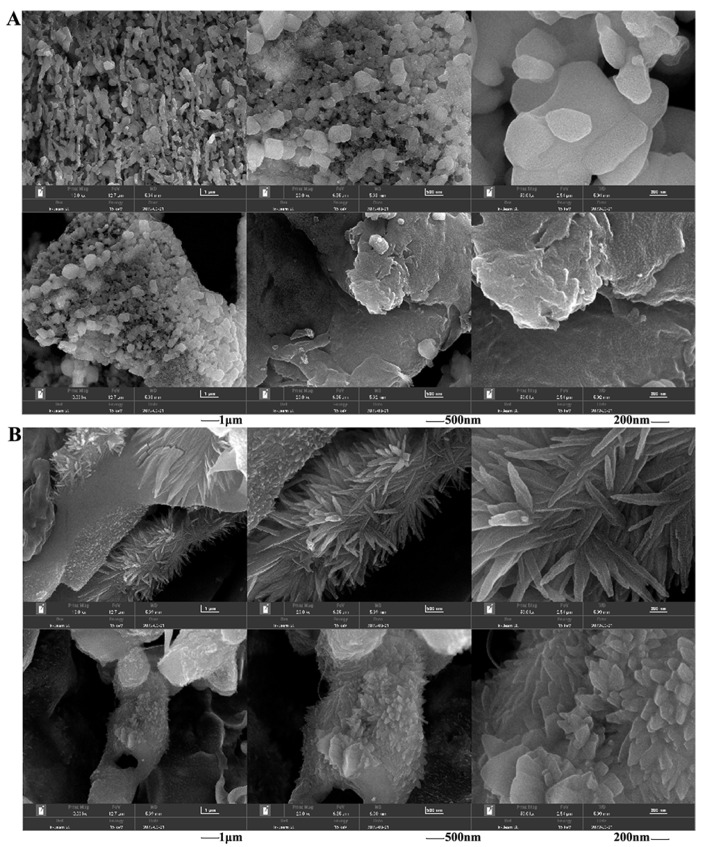
Morphological characteristics by scanning electron microscope of polysaccharides at different resolutions: (**A**), P1; (**B**), P2. The image scales are 1 μm, 500 nm and 200 nm.

**Figure 8 molecules-29-01238-f008:**
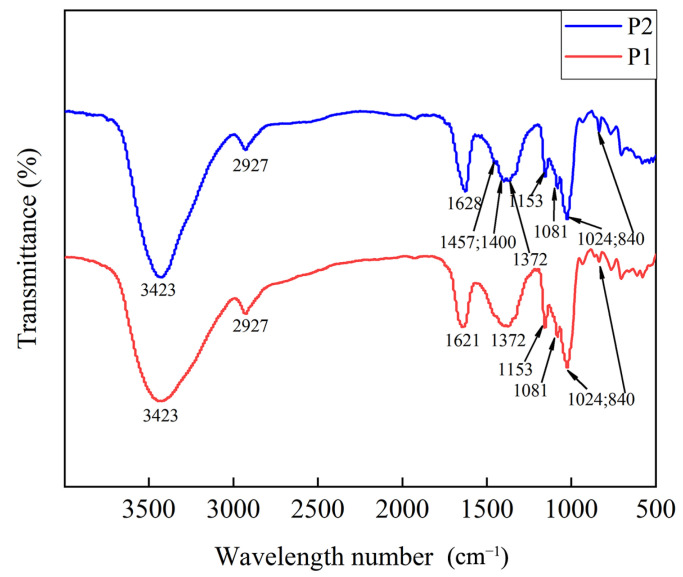
Infrared detection spectrum of P1 and P2.

**Figure 9 molecules-29-01238-f009:**
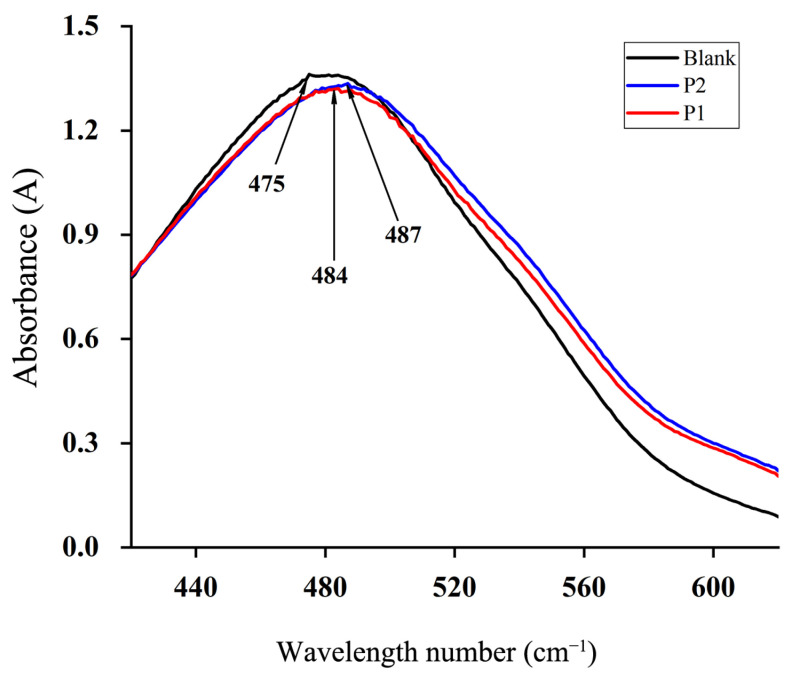
Distribution map of triple helix structure of P1 and P2.

**Table 1 molecules-29-01238-t001:** Response surface experimental results.

Run	X_1_: Time of Fermentation (d)	X_2_: Temperature of Fermentation (°C)	X_3_: Humidity of Fermentation (%)	Y: Yield (%)
1	−1	−1	0	3.73
2	−1	1	0	3.396
3	1	−1	0	3.345
4	1	1	0	2.908
5	0	−1	1	4.301
6	0	−1	−1	3.737
7	0	1	−1	3.633
8	0	1	1	3.809
9	−1	0	−1	3.431
10	1	0	−1	2.958
11	−1	0	1	3.624
12	1	0	1	3.257
13	0	0	0	4.434
14	0	0	0	4.415
15	0	0	0	4.522
16	0	0	0	4.456
17	0	0	0	4.414

**Table 2 molecules-29-01238-t002:** Molecular weights of P1 and P2.

Sample Name	Time Field (min)	Weight-Average Molecular Mass/u	Content (%)	Mr/u
P1	7.358–9.439	2.396 × 10^6^ ± 0.824	21.9	6.004 × 10^6^
	9.469–12.999	2.259 × 10^5^ ± 0.658	27.6	
	13.873–14.929	1.210 × 10^4^ ± 2.026	25.6	
	14.929–17.131	2.952 × 10^4^ ± 1.985	20.6	
	17.131–20.118	9.613 × 10^4^ ± 3.688	4.3	
P2	7.055–9.296	2.564 × 10^6^ ± 0.910	20.6	7.812 × 10^4^
	9.296–13.506	2.019 × 10^5^ ± 1.112	44.4	
	14.021–15.020	1.333 × 10^4^ ± 2.741	17.6	
	15.020–17.080	2.670 × 10^4^ ± 2.641	17.1	
	17.080–18.988	4.502 × 10^5^ ± 3.430	0.4	

**Table 3 molecules-29-01238-t003:** Design of response surface experiments.

Variate Conditions	Variable Level					Control Conditions
	1	2	3	4	5	
Time (d)	1	2	3	4	5	TEMP: 25 °C, HUM: 75%
Temperature (°C)	10	15	20	25	30	TIME: 3 d, HUM: 75%
Humidity (%)	55	65	75	85	95	TEMP: 25 °C, TIME: 3 d

## Data Availability

Data are contained within the article and Appendix A.

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
