# Peer review of "Biphasic Fermentation of *Trapa bispinosa* Shells by *Ganoderma sinense* and Characterization of Its Polysaccharides and Alcoholic Extract and Analysis of Their Bioactivity"

_molecules, 2024, doi:10.3390/molecules29061238_

Round 1

Reviewer 1 Report

Comments and Suggestions for Authors

The manuscript presents an interesting and novel approach to medical plant extract preparation and applications.  

The abstract is somewhat confusing. Please revise it; for example, the highlighted sentence in line 11 should be stated more clearly. The objective is difficult to ascertain. What was the hypothesis of this project?

The introduction is missing information.

Line 54, 63. Using abbreviations without introducing the concept before is confusing to the reader. Also, it suggests plagiarism (copy-paste)

Line 112: where is Table S1?

What type of statistics was applied to the data to prove the hypothesis

Line 259: Point 3 should be renamed to Results. There is no discussion within it

Line 302, 307. The numbers are reported with random precision. Significant figures rules should be applied throughout the results

Figure 3 is difficult to read; maybe the authors should separate it into two figures for better resolution. As an example, Figure 4 is well visualized.

Line 373: the reviewer does not have access to supplemented data, but regardless of this, the P1 and P2 spectra are the same.

Figure 6 Please indicate the magnification of the images via a clear indication of the sizes measured.

Lines 431-432, ‘acid cobalt staining results,’ are not presented. Did the authors make a typo when calling Congo red, cobalt staining?

Overall, the discussion is weak. There is minimal comparison to similar work or literature references. Please expand.

Comments on the Quality of English Language

English quality is acceptable

Author Response

Dear Reviewer,

We thank you very much for the comments and suggestions. We have carefully reviewed your comments and polished the text in the revised manuscript, below are answers to all of your questions.

  1. The abstract is somewhat confusing. Please revise it; for example, the highlighted sentence in line 11 should be stated more clearly. The objective is difficult to ascertain. What was the hypothesis of this project?

Response: The text has been revised in line 13-15 as “In this study, we optimized the solid-state fermentation process of Ganoderma sinense (GS) with TBs using a response surface experiment methodology to gain the fermented production with highest water extract rate and the DPPH free radical scavenging activity.”

  1. The introduction is missing information.

Response: The information has been added to show other research of TB’s bioactivity in line 13-15. “J.Xia et.al[13] showed the antioxidant, antiproliferative activity of the extract of Trapa bispinosa leaves, the alcoholic extract was absorbed by macroporous resin (D101) and eluted with different ratios of methanol-water and water, Fr.9 (eluted with 80% methanol) showed highest antiproliferative activity to HepG2, SSMC-7721, Hela and A549 cells, and showed high antioxidant activity to A549 cells. Ramsankar Sarkar[14] et.al's study mentioned that the polysaccharide of Trapa bispinosa showed high splenocyte, thymocyte and macrophage activation and antioxidant activity.”

  1. Line 54, 63. Using abbreviations without introducing the concept before is confusing to the reader. Also, it suggests plagiarism (copy-paste)

Response: The text has been revised in line 70-71 and 81. “Hasnat et al.[20] demonstrated that Ganoderma triterpenoids could effectively inhibit the expression of Cyclooxygenase-2(COX-2) and Interleukin-1 β(IL-1β) in the NF-кB pathway in mice with colitis, significantly alleviating the pathological condition of the mice.” “Li et al.[23] utilized biphasic fermentation technology to biotransform Marsdenia tenacis-sima(MT) with Ganoderma lucidum, identifying polysaccharides, saponins, organic acid alkaloids, and flavonoids of Ganoderma lucidum co-fermentation of MT by non-targeted metabolomics analysis and stoichiometric analysis.”

  1. Line 112: where is Table S1?

Response: Table 1 and Table S1 are misplaced, tables have been rearranged to the correct position. Table S1 was placed in “Support table.docx”.

  1. Line 259: Point 3 should be renamed to Results. There is no discussion within it

Response: Point 3 has been renamed to “Result” in line 279.

  1. Line 302, 307. The numbers are reported with random precision. Significant figures rules should be applied throughout the results

Response: Refer to other literature to change the text as below “The regression equation derived from regression analysis using Design-Expert software is as follows: Y = -95.0159 + 12.3334X1 + 1.8771X2 + 1.9424X3 - 0.0206X1X2 + 0.0106X1X3 - 0.0078X2X3 - 3.3119X12 - 0.0441X22 - 0.0121X32, with an R2 value of 0.9958.”; “The optimal extraction process comprised a fermentation duration of 1.93 days, a tem-perature of 14.1 ℃, and a humidity of 76.53%, resulting in a corresponding water extract yield of 4.52%.”; “The experiment was repeated three times to validate the process conditions, yielding a water extract yield of TBs at 4.36±0.02%.”; “Under these conditions, the DPPH radical scavenging rate of the aqueous extract ob-tained was 82.36±0.06%.”.

  1. Figure 3 is difficult to read; maybe the authors should separate it into two figures for better resolution. As an example, Figure 4 is well visualized.

Response: Figure 3 was reproduced with reference to Figure 4 and replaced in line 359, and rename as “Figure 4”.

  1. Line 373: the reviewer does not have access to supplemented data, but regardless of this, the P1 and P2 spectra are the same.

Response: Table S5 have been placed in “support table.docx”. In Table S5, the wavenumber 1640.94, 1457.52, 1399.56 and 773.61 appeared, and 1628.53 and 609.3 disappeared after polysaccharides fermentation. These changes of peaks were also showed in figure 8.

  1. Figure 6 Please indicate the magnification of the images via a clear indication of the sizes measured.

Response: Figure 6 was reproduced with indication of the sizes measured and replaced, and renamed as “Figure 7” in line 431.

  1. Lines 431-432, ‘acid cobalt staining results,’ are not presented. Did the authors make a typo when calling Congo red, cobalt staining?

Response: Text has been corrected as “acid Congo red” in line 477.

  1. Overall, the discussion is weak. There is minimal comparison to similar work or literature references. Please expand.

Response: Other researches about fungi fermentation and their bioactivity changes to RAW264.7 and other diseases have been added in line 525-539.

We sincerely appreciate your comments, if you have any questions, please contact us, hope you can accept our revision.

Thank you again for your comments and suggestions.

Best Wishes,

Xiaoyan Sun

Reviewer 2 Report

Comments and Suggestions for Authors

The manuscript provide a good optimization for the solid-state fermentation process of Ganoderma sinense (GS) with TBs using a response surface experiment. Authors prepared and characterized non-fermentation purified polysaccharides and fermentation purified polysaccharides. The Alcoholic extracts before (AE1) and after (AE2) fermentation were analyzed for active components such as polyphenols and flavonoids using UPLC-QTOF-MS/MS (Ultra Performance Liquid Chromatography-Quadrupole Time-of-Flight tandem mass spectrometry). Tables and figures present data efficiently. However, some concerns about statistical analyses to be addressed here.  Detailed statistical analyses should be provided in the materials and methods part including normality testing, parametric or nonparametric data, statistical analyses, statistical software, software version, ANOVA and posthoc to be included as well ….etc.

Author Response

Dear Reviewer,

We are grateful appreciate for your valuable comments. We totally understand the reviewer’s concern about statistical analyses. We have added the relevant contents of statistical analysis in line 280-284 in the revision.

We have carefully reviewed your comments and polished the manuscript again. If you have any questions to this revision, please contact us.

Once again, thank you very much for your comments and suggestions.

Best Regards,

Xiaoyan Sun

Reviewer 3 Report

Comments and Suggestions for Authors

Comments on the Quality of English Language

Author Response

Dear Reviewer,

We much appreciate your feedback on the manuscript. According to your recommendation, we endeavored to revise the relevant sections and made some changes to the manuscript. Below are answers to all of your questions.

Major comment:

  1. Title: Please rewrite the title by highlighting the optimized solid-state fermentation process and/or comparison of before and after fermantation, or gighlighting the differences obtained The Authors should more highlight the optimization and the characterization of obtained materials. The analysis of biological activitis was not the biggest and the most extensively studied component in this work.

Response: The title has been revised as “Biphasic fermentation of Trapa bispinosa shells by Ganoderma sinense and characterization of its polysaccharides and alco-holic extract and analysis of their bioactivity” in line 2-4.

  1. Abstract: please carefully revise and improve the abstract following the order: Background, Aim, Methodology, Results. Please briefly describe the impact of the tested P1 and P2 on the NO production and cytokine secretion. The biological effects are scarcely mentioned.

Response: The abstract has been revised followed the order: Background, Method, Result and Conclusion. The NO and cytokine secretion have been described in line 26-28.

  1. Introduction/Discussion: please describe in detail the anti-tumor, anti-oxidant and anti- /pro-inflammatory effects of Trapa Bispinosa or other herbs reported in previous works, especially focusing on macrophages/immune system if available.

Response: The bioactivities of Trapa bispinosa have been added in line 55-61. These researches were about anti-tumor, anti-oxidant and macrophage activation effects of Trapa bispinosa.

  1. Please revise the Materials&Methods and Results section headings. Make them more informative and be consistent with the language style, also revise the English laguage, e.g. 2.3, 2.4, 3.1, 3.2, 3.3, 3.5 (decide for gerund or relative clause in the whole work).

Response: The language of section headings have been revised in line 135 (2.3), 166 (2.4), 280 (3.1), 295 (3.2), 331 (3.3), 454 (3.6).

  1. Lane 256-258: The description of the qPCR data analysis is enigmatic. How did the Authors analyze the mRNA expression? What did the Authors mean by saying that the expression levels were compared? What house-keeping gene(s) was(were) used, and what method was used to calculate the relative expression? Please revise and correct.

Response: The analyzing method of qPCR was add in line 275-277, text as follow “The housekeeping gene used mouse ACTB gene (primer bought in Sangon, Shanghai, product id B662302), and the gene expression levels were analyzed by 2-△△Ct method.”

  1. A general comment to data presentation: please improve the graphical presentation (quality, size). It is ipossible to read e.g. graph legends, axis labelings. Please revise and rearrange the figures 1, 2,3,4,5.

Response: The figure have been rearranged, in line 293, 312, 360, 373, 416.

  1. Please revise and improve the language and scientific soundness for all figures and tables captions. Keep the style consistency.

Response: The language has been revised in line 132, 294-295, 313-314, 484.

  1. Please include the statistical analysis for Fig. 1 and Supplementary Fig. 1.

Response: Fig. 1 is in line 283-286, this part showed single factor optimization results indicated that when the fermentation humidity was 75%, the DPPH scavenging rate and the yield of water extract reached the highest. When the fermentation temperature was 15°C, the DPPH scavenging rate and the yield of water extract reached the highest. When the fermentation time was 2d, DPPH scavenging rate and yield of water extract reached the highest. Supplementary Fig.1 and 2 has been moved into main text and renamed as Fig.3, and this analysis was add in line 334-349.

  1. Lane 401: The refering to the results presented in the Fig. 3E should appear earlier in the text. The whole section 3.6. should be placed after the Section 3.3. Alternatively, the whole part of results reporting the biological effects of investigated materials could be placed as last section, and provided with the cytotoxicity analysis transferred from the supplementary part to the main text. Please carefully revise and improve this Results Section.

Response: The section 3.6 have been placed after section 3.3 in line 382-403, and renamed as “3.4. Changes at …”

  1. Discussion: Please discuss in detail the obtained results with other works reporting the fermentation process improvement strategies, and the biological activities of obtained materials. These issues were not addressed comprehensively.

a/ Please discuss in detail the changes in inflammatory cytokines, iNOS and NO, compare to other works, and give possible implications/benefits of this biological activity of the obtained materials.  

b/ Is there any possible medical use for the obtained extracts? What disease/pathological states? Are anti-oxidant, immuno regulatory properties desirable? Please discuss.

Response: The references of reporting the fermentation process changed the material s’ bioactivities were add in line 526-540. These researches include antitumor, antioxidant, immunomodulatory effects, and the objects of researches include RAW264.7 cells and some disease.

Minor comment

  1. Lane 9: please replace “antioxidative and anti-tumor immune-regulating effects” with “anti-oxidant, anti-tumor, and immunomodulatory effects”.

Response: Has replace in line 10.

  1. Lane 11: please replace “experiment” with “methodology”.

Response: Has replace in line 14.

  1. Lane 16: please replace with “macrophages (RAW 264.7) were”

Response: Has replace in line 20.

  1. Lane 32: plase replace “anit-oxidation” with “anti-oxidant”.

Response: Has replace in line 42.

  1. Lane 33: please replace “regulation” with “regulatory”.

Response: Has replace in line 43.

  1. Lane 55: please be specific with “condition”, e.g. “pathological condition”.

Response: Has replace in line 72.

  1. Lane 62: please be speciic, provide examples for these specific pharmacological activities.

Response: Has add in line 79-80 as “This process provides nutrients for bacteria and simultaneously alters the Chinese medicinal base material through the fungal enzyme system, resulting in complex bac-teria with specific pharmacological activities, such as anti-inflammatory and anti-tumor activity.”

  1. Lane 67: please replace “this” with “This process”.

Response: Has replace in line 85.

  1. Lane 68: What drug did the Authors mean? Please be more specific, provide more details.

Response: Has replace in line 85 into “original medicine”. This means the plant material (Marsdenia tenacissima) used in the reference article.

  1. Lane 83:Please replace “Sigma” with “Sigma Aldrich”, “DEME” with “DMEM”.

Response: Has replace in line 101.

  1. Lane 87” please replace “Synthase” with “Synthase Activity”.

Response: Has replace in line 105.

  1. Lane 86: please give the cat. no. or bacterial strain origin of the LPS used.

Response: The cat. No. has add in line 104 as “lipopolysaccharide (LPS) isolated from Escherichia coli 0111:B4”.

  1. Lane 90: please replace “isoplus” with “isolation”.

Response: RNAiso plus is the name of the RNA isolation kit of TAKARA. Name have been correct in line 108.

  1. Lane 104: The text do not refer to the Table 1 mentioned. Please revise.

Response: The table 1 and table S1 were misplaced, these table have been replaced in correct place in line 132 and support table file.

  1. Lane 114: Please revise the table heading, e.g. Design of Response Surface Experiments.

Response: Has revised in support table file as “Table S1. Design of Response Surface Experiments”.

  1. Lane 121: please provide the reference for the method.

Response: Has provide the reference in reference article 25.

17, Lane 122: please replace “20-minute” with “20 minutes”.

Response: Has replace in line 141-142 as “… followed by 20 minutes standing period before…”

  1. Lane 125, 179, 183, etc.: please replace the centrifugation parameter r/min with xg to easily repeat the centrifugation procedure by others.

Response: Has add xg after the origin text in line 138, 145, 202, 205 as “10000r/min(9168xg)”, “4500r/min(1856xg)”, “12000 r/min(13201xg)”, “13000 r/min(15493xg)”.

  1. Lane 127” please replace “20ml” with “20ml of”.

Response: Has replace in line 147.

  1. Lane 160-162: please revise and rewrite the sentence.

Response: Has revised in line 181-182 as “The RAW264.7 cells’ secretion of the blank group and the positive group were compared to de-termine the changes of immunomodulatory activity before and after fermentation.”

  1. Lane 165: please rewrite the sentence, e.g. “exhibiting no cytotoxic effects”.

  1. Lane 169: What did the Authors mean by “drug”? Please be more specific, e.g. the material, extract, etc. No drug was used in this study.

Response: “drug treatment” has been replaced into “extract treatment” in line 189.

  1. Lane 173-175: please rewrite the sentence. Secretion of what?

Response: The text has been wroten in line 195-197 as “The RAW264.7 cells’ secretion of the blank group and the LPS positive control group were compared to determine the changes of antioxidant activity before and after fer-mentation.”

  1. Lane 200: please provide the reference for the Congo red method.

Response: The reference for the Congo red method was add in line 221 and reference article 26.

  1. Lane 212: please revise the sentence, e.g., replace with “determine the composition of AE1 and AE2 samples”.

Response: Has replaced the sentence in line 234.

  1. Lane 223-224: please remove “Mass…. As follows:”.

Response: The text has been removed.

  1. Lane 234: it should be highlighted that it was mRNA expression analysis, not proteins.

Response: The title has been revised as “2.7. Analysis of the mRNA expression levels on NF-κB pathway and Nrf2/Keap1-ARE pathway” to highlight the mRNA.

  1. Lane 236-239: please rewrite the sentences, e.g., “To investigate the changes in the expression levels of NF-κB pathway-related mRNA, four experimental groups were established: the control group…”. Similar comment for Lanes 242-243.

Response: The text has been revised in line 256-257, 261-262 as “To investigate the changes in the expression levels of NF-κB pathway-related genes, four experimental groups were established: …”, “Similarly, to investigate the changes in the expression levels of Nrf2/Keap1-ARE pathway-related genes, four experimental groups were analyzed: …”.

  1. Lane 240, 246: please remove “cell solution”.

Response: Have been removed in line 259, 264.

  1. Lane 243: please replace “mRNA” with “genes”, “set up” with “analyzed”.

Response: Have been replaced in line 262 as “Similarly, to investigate the changes in the expression levels of Nrf2/Keap1-ARE pathway-related genes, four experimental groups were analyzed: …”.

  1. Lane 249: What did the Author mean by “drug”? Please revise and correct as previousley commented.

Response: Replaced “drug” with “extract” in line 268.

  1. Lane 251: please be more specific with “micro-UV”, revise.

Response: Replaced “micro-UV” with “Ultra-micro UV-visible spectrophotometer” in line 270-271.

  1. Lane 253: please provide the supplier details for the RT kit.

Response: The supplier of the RT kit is TAKARA. Detail has been added into line 272.

  1. Lane 259: Please remove “and discussion”.

Response: Has removed “ and discussion” in line 280.

  1. Lane 326: please rewrite the text regarding MDA, and define MDA here.

Response: Text has been revised in line 368-371 as “As shown in Figure 5, following treatment with AE1 and AE2, T-AOC and SOD showed a significant increase, while Malondialdehyde (MDA) secretion exhibited a significant decrease. MDA is one of the final products of polyunsaturated fatty acids peroxidate by H2O2 in the cell.”.

  1. Lane 329: please replace “MAD” with “MDA”.

Response: Miss word has been replaced in line 374.

  1. Lane 336: please rewrite the sentence, e.g. “…alcoholic extracts was performed”.

Response: The sentence has been revised in line 380-381 as “To investigate the causes of these functional changes, the structure of polysac-cha-rides and the components of alcoholic extract was performed.”.

  1. Lane 351: please rewrite the sentence, e.g. “The content distribution of fractions with different weight-average molecular masses in P1 was relatively homogeneous (Table 3).

Response: The sentence has been revised in line 418-419 as “The content distribution of fractions with different weight-average molecular masses in P1 was relatively homogeneous (Table 3).”

  1. Lane 365: please replace “image” with “images”.

Response: The word has been replaced in line 433.

  1. Lane 395 and others: please write the name of compounds, metabilites with lower case letter.

Response: The names have been revised in line 463 as “A comparison between AE1 and AE2 revealed a reduction in polyphenolic compounds, such as [6]-gingerol, (-)-gallocatechin, and cynaroside, while flavonoids showed that the addition of myricetin and maltol and a reduction in alkaloid alkaloids like DL-tyrosine.”.

  1. Lane 396, 397: please improve the scientific language, revise “saw”, “experienced”.

Response: The language has been revised in line 464, 465 as “A comparison between AE1 and AE2 revealed a reduction in polyphenolic compounds, such as [6]-gingerol, (-)-gallocatechin, and cynaroside, while flavonoids showed that the addition of myricetin and maltol and a reduction in alkaloid alkaloids like DL-tyrosine. Triterpenoids in AE2 showed that the addition of ganoderol A, linderalactone, and lamiide, with a reduction in artemisinin.”.

  1. Lane 437: please replae “functions” with “properties” or “activities”.

Response: The word has been replaced in line 484 with “activities”.

  1. Lane 452, 469: evaluation criteria for what/to assess what? Please be more specific.

Response: The text has been revised in line 498-500 as “In this study, to determine the changes of immunomodulatory effects of the extract before and after fermentation, NO, iNOS, TNF-α, IL-2, and IL-10 were employed as evaluation criteria.”

  1. Lane 461: please define the abbreviation “T-AOC”.

Response: The text has been revised in line 509 as “Total antioxidant capacity (T-AOC) represents the collective ability of antioxidant substances in the body to combat oxidative damage [37].”.

  1. Lane 485: please revplace “in biological” with “in the biological”.

Response: The word has been replaced in line 548.

  1. Lane 244 and others to 412: please define “VC”. Include the term in the Materials part and Figure captions to avoid confusion about the group treatment design.

Response: The word “VC” has been replaced with “L-Ascorbic acid (VC)”.

  1. The whole manuscript: please revise the terms pre-fermentation and before fermentation; alcohol and alcoholic extracts, they are used interchangeably. Please revise and decide for one style to avoid confusion.

Response: The text has been revised into “before and after fermentation”, “alcoholic extract”.

  1. Supplementary File: Table S6; Please revise “Contencontent” in the Lane 1 of the Table.

Response: The word has been revised into “Content of Relative Percentage (%)”.

  1. Supplementary File: Table S6; please revise the “Compound” column (revise column widths), some names are split into two lanes resulting in uneven text position, e.g. Ethyl 4-hydroxybenzoate, 4'-Prenyloxyresveratrol, 3',4'-Anhydrovinblastine, 1,3,6-Tri-O-galloyl-β-D-glucose, etc

Response: The column widths have been revised to fit the words.

We sincerely appreciate your kind efforts and hope that the revision will be accepted. Should you have any questions, please contact us without hesitation. 

Once again, thank you for the precious time you spent making constructive comments.

Best Wishes,

Xiaoyan Sun

Round 2

Reviewer 1 Report

Comments and Suggestions for Authors

Dear Authors,

I congratulate you on the improvemtns done to the manuscript.

There are some editing issues, such as italicization of scientific names and too many spaces between words, but otherwise, the manuscript is ready.